# The Application of Synthetic Flavors in Zebrafish (*Danio rerio*) Rearing with Emphasis on Attractive Ones: Effects on Fish Development, Welfare, and Appetite

**DOI:** 10.3390/ani13213368

**Published:** 2023-10-30

**Authors:** Federico Conti, Matteo Zarantoniello, Matteo Antonucci, Nico Cattaneo, Mirko Rattin, Gaia De Russi, Giulia Secci, Tyrone Lucon-Xiccato, Adja Cristina Lira de Medeiros, Ike Olivotto

**Affiliations:** 1Department of Life and Environmental Sciences, Marche Polytechnic University, 60131 Ancona, Italy; f.conti@pm.univpm.it (F.C.); n.cattaneo@pm.univpm.it (N.C.); mirko.rattin@gmail.com (M.R.); 2Independent Researcher, Via Pola 18, 64014 Martinsicuro, Italy; matteo.antonucci.89@outlook.com; 3Department of Life Sciences and Biotechnology, University of Ferrara, 44121 Ferrara, Italy; gaia.derussi@unife.it (G.D.R.); tyrone.luconxiccato@unife.it (T.L.-X.); 4Department of Agriculture, Food, Environment and Forestry, University of Florence, 50144 Firenze, Italy; giulia.secci@unifi.it (G.S.); adjacristina.lirademedeiros@unifi.it (A.C.L.d.M.)

**Keywords:** feed attractant, zebrafish development, histology, feed intake, growth factors

## Abstract

**Simple Summary:**

Low aquafeed palatability can adversely affect fish growth and welfare, often resulting in feed waste. Conventional feed attractants, which are regularly included in aquafeeds to improve palatability, rely on natural marine resources. The present study aimed to identify cost-effective synthetic flavors to be included as innovative feed attractants in the zebrafish (*Danio rerio*) diet. A comprehensive overview on the effects of the tested flavors—two attractive and one repulsive—on fish growth and development, welfare, and appetite regulation was analyzed, comparing zebrafish larval and juvenile stages. Both diets containing attractive flavors enhanced zebrafish feed intake and growth without affecting fish welfare. Maximizing feed intake is a critical point for the aquafeed production sector to reduce feed wastage and minimize the economic impact.

**Abstract:**

The aim of the present study was to test synthetic flavors as potential feed attractants in zebrafish (*Danio rerio*) during early development. Six experimental groups were set up in triplicate: (i) a CTRL group fed a zebrafish commercial diet; (ii) a PG group fed a control diet added with Propylene Glycol (PG); (iii) A1^+^ and A2^+^ groups fed a control diet added with 1% of the two attractive flavors (A1^+^ cheese odor made by mixing Propylene Glycol (PG) with the aromatic chemicals trimethyamine, 2-acetylpyrazine, 2-acetylpyridine, and dimethyl sulfide; and A2^+^ caramel odor, made of PG mixed with the aromatic chemicals vanillin, maltol, cyclotene, acetoin, butyric acid, and capric acid with traces of both gamma-octalactone and gamma-esalactone) or the repulsive flavor (A^−^ coconut odor, made by mixing PG with the aromatic chemicals gamma-eptalactone, gamma-nonalactone, delta-esalactone, and vanillin with trace of both delta-octalactone and maltol), respectively; (iv) an ROT group fed the two attractive diets, each administered singularly in a weekly rotation scheme. All the tested synthetic flavors did not affect the overall health of larval and juvenile fish and promoted growth. Due to the longer exposure time, results obtained from the juvenile stage provided a clearer picture of the fish responses: zebrafish fed both attractive diets showed higher appetite stimulus, feed ingestion, and growth, while the brain dopaminergic activity suggested the A2^+^ diet as the most valuable solution for its long-lasting effect over the whole experiment (60-day feeding trial, from larvae to adults). The present study provided important results about the possible use of attractive synthetic flavors for aquafeed production, opening new sustainable and more economically valuable opportunities for the aquaculture sector.

## 1. Introduction

In the past few decades, aquaculture has significantly increased in terms of food production. For instance, from 2010 to 2020, this sector globally grew at an average annual rate of 4.6%, reaching a record of 122.6 million tons in 2020, including 87.5 million tons of aquatic animals worth USD 264.8 billion [1]. This is largely due to the food demand of an expanding world population, which is estimated to reach 9 billion people by 2050. Human fish consumption has more than doubled in the past 50 years, reaching 20.2 kg per capita in 2020 [1]. However, to ensure further development of the aquaculture sector, proper resource management is required to promote sustainability and circularity [2], with emphasis on the aquafeed production sector, which still relies on the use of conventional marine- and plant-derived raw materials [3]. Fish meal (FM) and fish oil (FO) represent the ideal ingredients for aquafeeds, ensuring proper fish growth and welfare [4], but due to their increasing costs and ecological drawbacks, they have been gradually replaced, over the last few decades, by plant-derived ones [5]. However, the use of plant protein sources like soybean or corn meals for fish feed formulations, besides leading to food vs. feed competition, presents several negative side effects on fish growth and welfare, especially for carnivorous species [6,7]. The presence of nondigestible carbohydrates and antinutritional factors can adversely affect nutrient digestibility/bioavailability, and consequently, fish health status [8], in addition to often being poorly palatable [3]. Part of these problems has been addressed by (i) improving the processing methods of plant-derived ingredients (protein hydrolysates; inclusion of feed supplements such as butyrate) [9,10]; (ii) replacing plant-derived ingredients with promising and more sustainable alternatives of animal origin like poultry by-product, insect, and crayfish meals or proteins derived from microbial biomasses, including fungi, microalgae, and cyanobacteria [11,12,13,14].

However, the production of more sustainable diets often results in lower diet palatability, affecting fish growth, water quality, and farm economics [15]. In fact, it is estimated that a considerable portion of feed administered in fish farms is not ingested by fish, ending in wastewater outflows and contributing to eutrophication of aquatic ecosystem and producing economic losses [16,17,18].

Nowadays, a class of feed additives, known as feed attractants, are regularly included in aquafeeds to improve nutrient-sensing of fish [19]. Chemical stimuli originated from highly palatable feed can influence both feed-seeking and ingestion, by activating the gustatory and olfactory senses of fish [20]. In vertebrates, feeding is a complex mechanism controlled by a fine brain–gut crosstalk mediated by neurohormonal signals [21]. The appetite stimulus is induced by different orexigenic signals, among which ghrelin and neuropeptide Y (NPY) have been widely demonstrated to play an essential role in stimulating feeding in teleost [22]. Ghrelin is directly produced by the entero-endocrine cells of the gut tract and possesses a strong orexigenic role promoting, at a central level, the expression of *npy* [23,24]. Conversely, a plethora of anorexigenic signals both produced by central and peripheral tissues play a major role in suppressing feeding [22]. In this context, leptin decreases the feeding intake by inhibiting the orexigenic signals and stimulating the anorexigenic pro-opiomelanocortin (POMC) and cocaine- and amphetamine-regulated transcript (CART) system at central level [25] and induces the mobilization of energy reserves by stimulating lipolysis [23,26].

Fish feeding behavior and appetite sensation are also closely related to the brain reward system, in which dopamine is the main neurotransmitter involved in the hedonic appetite regulation, that increases motivation after the ingestion of palatable feed [26,27]. Thus, improving the organoleptic properties of fish diets through the use of specific feed attractants represents a critical aspect to elicit an optimal feeding response by fish in terms of time and feed intake [10]. Nowadays, feed attractants for aquafeeds are mainly represented by natural-derived ingredients [28]. Meals obtained from shrimp, anchovy, and squid are well-recognized feed attractants, regularly included in fish diets to improve palatability [29,30]. On this regard, Kader et al. [31] tested the effects of fish soluble (*Katsuwonus pelamis* by-product), krill meal, squid meal, and crystalline amino acids as feed attractants in a plant-based diets on red seabream, *Pagrus major*, concluding that all of these ingredients acted as feed attractants. Additionally, Nagel et al. [30] investigated the potential of blue mussel meal as feed attractant in a plant-based diets for turbot (*Psetta maxima*), observing similar results. All of these solutions play a positive role as feed attractants, but their production still depends on marine resources, posing further unsustainability issues for the aquafeed production sector [32]. In addition, due to their natural origin, these ingredients present fluctuations in availability and their attractive effect is highly variable depending on the raw material composition, freshness, and processing methods [33].

The current alternative solutions that have been tested are represented by a limited number of molecules, such as a mixture of free amino acids (e.g., glycine, L-alanine), nucleosides and nucleotides, or substances such as betaine and taurine, which can be used as feed attractants [34,35,36]. Marine extracts are rich in all these compounds, although amino acids appear to be the main class of molecules that induce feed-seeking behavior in fish [37]. Amino acids have been widely studied for their attractive properties, although the limitations and disadvantages of their use as feed attractants are well known [38]. It is noteworthy that free amino acids exhibit different attractive effects depending both on the fish species and the life cycle stage considered, thus leading to controversial results. Furthermore, the attractive effect depends also on the amino acid used as well as on its concentration and interaction with other molecules present in the diets. Finally, there is a lack of data in the scientific literature on the identification of optimal dosage for different fish species, and anyway, the application of amino acids as feed attractants is less effective than the above mentioned marine-derived ingredients [36,38].

Therefore, a novel, standardized, and sustainable alternative to natural feed attractants may be represented by synthetic flavors, obtained through standardized processes. The available literature provides only few specific studies on this topic [28], which instead can represent a great opportunity for the aquaculture sector to satisfy the organoleptic requirements of fish [39]. Indeed, it is well known that in the food industry, a limited number of molecules allow us to replicate specific flavors [40]. A synthetic flavor is composed by a mixture of aromatic chemicals such as esters, terpenes, aldehydes, phenols, lactones, fatty acids, and sulfur compounds [41], recognized as safe for use by the American Flavor and Extracts Manufacturers Association (FEMA). These substances are combined in a liquid blend with a functional solvent (e.g., water, triacetin, propylene glycol) to impart or improve a particular flavor (e.g., banana, cherry, chocolate) to different products [39,41]. In this way, thousands of different food-related flavors can be faithfully reproduced through sustainable and cost-effective processes [42].

In this context, the aim of the present study was to identify different synthetic flavors that are widely used for human food consumption, and to test them during zebrafish (*Danio rerio*) culture to assess their potential role as feed attractants. Zebrafish represents a well-validated model for aquaculture nutritional studies and allows us to investigate possible dietary effects on all its life stages in a relatively short time [43,44,45,46]. A set of positive and negative commercial synthetic flavors were initially identified through a behavioral preference test and secondly added to a specific zebrafish diet which was administered to fish from the larval to the juvenile stage. At the end of both larval and juvenile phases, biological samples were collected and analyzed through a multidisciplinary approach to obtain a comprehensive overview of the fish physiological responses to the synthetic flavors here tested.

## 2. Materials and Methods

### 2.1. Ethics

All the experimental procedures involving animals conducted in the present study were performed in accordance with the Italian legislation on experimental animals and were approved (n° 2-1/12/22) by the Ethics Committee of the Marche Polytechnic University (Ancona, Italy) and the Italian Ministry of Health (Aut N° 453/2023-PR). An anesthetic (MS222 1 g/L; Merck KGaA, Darmstad, Germany), was used to minimize the suffering of the animals.

### 2.2. Synthetic Flavors

Forty commercial flavors were produced and blindly provided by To Be Pharma S.r.l. (Via Vibrata 127, 64016, S. Egidio alla Vibrata, Teramo, Italy). All the 40 synthetic flavors tested were composed by the same basic solvent (1,2-propanediol, Propylene Glycol) and a mixture of different flavoring chemicals, produced through a standardized process and widely used for human consumption [39,41]; Flavor and Extracts Manufacturers Association—FEMA. The use of Propylene Glycol has also been classified as safe for use by Food and Drug Administration (FDA; Agency for Toxic Substances and Disease Registry, 2007).

As regards the selected flavors for the present study, the F25 one was composed of Propylene Glycol (PG) as solvent mixed with the following aromatic chemicals: trimethylamine, 2-acetylpyrazine, 2-acetylpyridine, and dimethyl sulfide. Differently, the F35 flavor was composed by PG as solvent, mixed with the aromatic chemicals vanillin, maltol, cyclotene, acetoin, butyric acid, and capric acid with traces of both gamma-octalactone and gamma-esalactone. Finally, the F32 flavor was composed by PG, as solvent mixed with the aromatic chemicals gamma-eptalactone, gamma-nonalactone, delta-esalactone, and vanillin with traces of both delta-octalactone and maltol. The final odors of F25, F35, and F32 flavors were cheese, caramel, and coconut, respectively. Due to intellectual property protection, the composition of each single tested flavor in the present study is reported as relative abundance of each single chemical (first one more abundant, last one less abundant). 

All products tested were produced with nontoxic food flavorings developed for human consumption in agreement with the current sector-specific legislation (Regulation (EC) No 1334/2008 and 1333/2008) and are safe for humans and animals, as reported from the technical data sheet and the Material Safety Data Sheet (MSDS).

### 2.3. Preliminary Behavioral Preference Test

All 40 synthetic flavors were tested through a behavioral preference assay conducted on zebrafish larvae following a previously published protocol developed for this species [47]. This preliminary study was performed to identify the flavors determining a significant behavioral response (i.e., attraction or avoidance) to be used in the following analyses of the study. 

Each flavor was presented to 20 individual zebrafish larvae (N = 800 larvae in total; N = 20 replicates per flavor). The larvae were obtained through a standard breeding protocol from groups of adult zebrafish of the AB strain. In detail, the adults were kept in spawning tanks with a bottom grid that permitted to collect the eggs upon fertilization and avoid cannibalism. Eggs were collected from multiple spawns of different adult zebrafish. After collection, the eggs were maintained in small groups in Petri dishes until hatching, and then moved into a Tecniplast system (Varese, Italy) with continuous water circulation and filtration until the behavioral test. 

For the testing, subjects from the pool of zebrafish larvae (21 days postfertilization; dpf) were randomly selected and were individually moved into 8 × 24 cm white plastic tanks, filled with 0.5 L of clean water. Multiple experimental tanks were used to simultaneously test multiple subjects. After 30 min of acclimatization, in each tank, a cellulose sponge (1 cm^3^) soaked with one of the 40 flavors was introduced. The specific flavor was randomly assigned to each subject. The sponge with the stimulus was placed in correspondence of one of the shorter sides of the testing tank, kept in place close to the bottom by a small pin. The flavors were diluted 1:100 with distilled water. Additionally, a second sponge soaked with distilled water only was used as control and located on the opposite side of the flavor-soaked sponge. The behavior of the subject was video-recorded thanks to cameras placed above the tanks (Sony LEGRIA HFR38) using tripods. The test duration was 15 min, during which each subject was left free to swim. After each test, the tanks were completely washed with distilled water and refilled with clean water before testing another subject.

The video recordings were successively analyzed using BORIS software (ver. 7.12.2) [47]. This allowed us to calculate the time spent by each larva close to the stimulus (flavored sponge) or close to the control sponge, for each minute of the test. As detection threshold, an area of 1/3 of the experimental tank for each stimulus (8 × 8 cm) was used. These data were managed to compute an index of percentage preference for the flavor controlled for the control sponge as follows: (time close to the flavor/(time close to the flavor + time close to the control sponge)) × 100. This index allowed us to exclude time spent by the subject far from both stimuli (i.e., in the center of the experimental tank). The index ranged from 0 to 100%, assuming different values according to the preference of the subjects: it showed values higher than 50% when a subject spent most of the time close to the flavor; conversely, the index had values smaller than 50% when a subject spent most of the time close to the control sponge with no flavor. In case of no preference, the value of the index was close to 50%. The index was calculated in two-time blocks of 0–5 and 0–15 min, respectively, because this preference usually varies across the testing time [47].

### 2.4. Production of Experimental Diets

Starting from a commercial diet for zebrafish (Zebrafeed; Sparos LDA, Olhão, Portugal), used as control (CTRL diet) (for composition, please see [48]), four experimental diets were prepared as follows: (i) PG diet obtained by adding 1% (*w*/*w*) of Propylene Glycol (the basic solvent of the flavors; PG) to the control diet; (ii) A1^+^ diet obtained by adding 1% (*w*/*w*) of F25 as attractive/positive flavor to the control diet; (iii) A2^+^ diet obtained by adding 1% (*w*/*w*) of F35 as attractive/positive flavor to the control diet; (iv) A^−^ diet obtained by adding 1% (*w*/*w*) of F32 as repulsive/negative flavor to the control diet. Flavors and PG were added daily to the right amount of the control diet at 1% (*w*/*w*) using a micropipette and vigorously mixed to guarantee a homogenous distribution in the feed.

### 2.5. Experimental Design

Zebrafish AB strain adults fed on a commercial diet for zebrafish (Zebrafeed; Sparos LDA, Olhão, Portugal) were laboratory-spawned and the obtained embryos were collected and maintained according to Cattaneo et al. [49]. After 48 hrs of embryo development, a total number of 9000 live and developing embryos were selected under a stereomicroscope, collected, and then randomly divided into six experimental groups (3 tanks per experimental group): (i) CTRL group, zebrafish fed the control diet); (ii) PG group, zebrafish fed control diet added with 1% of Propylene Glycol (PG diet); (iii) A1^+^ group, zebrafish fed control diet added with 1% of positive/attractive flavor (diet A1^+^); (iv) A2^+^ group, zebrafish fed control diet added with 1% of positive/attractive flavor (diet A2^+^); (v) A^−^ group, zebrafish fed control diet added with 1% of the negative/repulsive flavor (A^−^ diet); (vi) ROT group, zebrafish fed the two attractive diets (A1^+^ and A2^+^), each administered singularly in a weekly rotation scheme. This last group was included in the experimental design since it has been demonstrated that teleost, in response to a repeated stimulation, may develop an olfactory receptors adaptation [50]. Zebrafish larvae were initially reared in 20 L tanks (500 larvae per tank, 3 tanks per experimental group) with the same water conditions of the broodstock’s tanks and provided by a dripping system to ensure a proper water change and black panels placed on the sides of each tank to reduce light reflection [51]. After 21 days postfertilization (dpf), zebrafish from each tank were gently moved to other tanks (100 L, 3 tanks per experimental group) provided with mechanical and biological filtration (Panaque, Capranica, Italy).

Starting from day 5 dpf to the end of the trial (60 dpf), zebrafish were fed the experimental diets, tanks were cleaned and dead specimens collected and counted, according to Rashidian et al. [52] and Zarantoniello et al. [48]. 

The required number of fish per tank was sampled, after a lethal dose of MS222 (1 g/L, Merck KGaA, Darmstad, Germany), at 21 dpf to evaluate dietary effect of flavors administration during the larval development. The remaining fish were sampled at 60 dpf, after being euthanized with a lethal dose of MS222 (1 g/L, Merck KGaA), to evaluate the long-term dietary effect of flavor administration on juvenile stage.

### 2.6. Biometry

Ten just-hatched larvae per tank (30 per dietary group) were randomly collected at 3 dpf to measure the initial body weight (IBW). Then, 20 zebrafish larvae and 20 zebrafish juveniles (60 larvae and 60 juveniles per experimental group) were randomly collected from each tank at 21 and 60 dpf, respectively, to measure the final body weight (FBW). The specific growth rate (SGR) was calculated as follows:
(1)SGR = [(ln FBW − ln IBW)/t] × 100
in which t represents the number of feeding days (17 and 57 for larvae and juveniles, respectively). The survival rate was calculated at 21 and 60 dpf for both the developmental stages, by deleting the dead specimens to the initial number of fish.

### 2.7. Feed Intake

The experiment was conducted only in the juvenile stage since, for the larval stage, the sizes of both pellets and fish were too small to have an accurate result. Fifteen zebrafish juveniles per tank (45 per experimental group) were acclimatized for one week prior to the feed intake experiment and were fed the experimental diets, providing a pre-weighed quantity of feed corresponding to 3% of their body weight. During the feed intake experiment, eventually, uneaten feed was recovered 15 min post-administration by siphoning. The recovered uneaten feed was dried in an oven overnight at 40 °C for quantification. The duration of the test was selected according to the maximum duration of the behavioral preference test for flavor selection (15 min). 

### 2.8. Histological Analysis

Five whole larvae per tank (15 per experimental group) were collected at 21 dpf, and liver and whole intestine samples from 5 juveniles per tank (15 per experimental group) were collected at 60 dpf according to Cattaneo et al. [49]. Samples were fixed by immersion in Bouin’s solution (Merck KGaA) for 24 h at 4 °C and then washed three times with 70% ethanol for 15 min and maintained in a new 70% ethanol solution at 4 °C. Using graded ethanol solutions (80, 95, and 100%) samples were dehydrated, then washed with xylene (Bio-Optica, Milano, Italy), and finally embedded in paraffin (Bio-Optica). Using a microtome (Leica RM RTS, Nussloch, Germany), sections of 5 µm were cut from the solidified paraffin blocks. Sections were stained with Mayer hematoxylin and eosin Y (Merck KGaA) to estimate possible alterations in tissue architecture and presence of inflammations in both the intestinal tract and the liver.

The evaluation of histological indexes in the intestine was performed on three transversal sections per fish (15 whole larvae and 15 intestines from juveniles per experimental group) collected at 50 μm intervals according to [49,53]. Specifically, for the morphometric evaluation of mucosal fold height, all the undamaged and non-oblique folds were measured using ZEN 2.3 software (Zeiss, Oberkochen, Germany). Regarding the analysis of histopathological indexes, scores were assigned as follows: (i) inflammatory influx: + = scarce lymphocyte infiltration, ++ = moderate lymphocyte infiltration, +++ = diffused lymphocyte infiltration; (ii) mucosal folds fusion: + = 0–3 observations per section, ++ = 3–10 observations per section, +++ ≥ 10 observations per section.

### 2.9. Molecular Analyses

Briefly, total RNA extraction and cDNA synthesis from 3 larvae per tank (in triplicate, 9 larvae per group) collected at 21 dpf, and from brain, liver, and whole intestine samples from 3 juveniles per tank (in triplicate, 9 brains, 9 livers, and 9 intestines per group), collected at 60 dpf, were performed according to Cattaneo et al. [49] and Olivotto et al. [54].

Real-time quantitative PCR (qPCR) reactions were performed in an iQ5 iCycler thermal cycler (Bio-Rad, Hercules, CA, USA) on a 96-well plate according to Randazzo et al. [55], including a nonreverse transcription control for each reaction to exclude possible contamination by genomic DNA. For each sample, reactions were set according to Cattaneo et al. [49]. The thermal profile was 3 min at 95 °C, followed by 45 cycles of 20 s at 95 °C, 20 s at the melting temperature for each primer, as reported in Table 1, and 20 s at 72 °C. Annealing temperatures for each primer were optimized through temperature gradient assays, and primer specificities were assessed with the absence of primer–dimer formation and dissociation curves. Additionally, for each primer, efficiencies were evaluated with a mix of cDNA from CTRL group at different concentrations (1:1, 1:10, 1:100, 1:1000); for all the primers tested, efficiency was around 90%, with an R^2^ that ranged from 0.995 to 0.998. At the end of each cycle, the fluorescence was monitored, and the melting curve analyses showed one single peak for every product. Absence of contamination and no peaks were found in the two no-template controls (NTCs) for each reaction. Relative mRNA abundance was used to quantify the gene expression using, as housekeeping genes, the ribosomal protein L13 (*rpl13*) and actin-related protein 2/3 complex subunit 1a (*arpc1a*) to standardize the results. The relative quantification of genes involved in fish growth (insulin-like growth factor, *igf1*; myostatin, *mstnb*), appetite regulation (ghrelin, *ghrl*; neuropeptide Y, *npy*; and leptin, *lepa*), brain reward system (dopamine receptor D1b, *drd1b*; dopamine receptor D2a, *drd2a*; dopamine receptor D3, *drd3*), stress response (glucocorticoid receptor, *nr3c1*), and immune response (interleukin 1β, *il1b*; interleukin 10, *il10*; lipopolysaccharide-induced TNF factor, *litaf*) was performed on samples from pool of whole larvae or from target organs considering juveniles. Particularly, (i) *igf1*, *mstnb*, *nr3c1*, and *lepa* were tested in liver samples; (ii) *ghrl*, *il1b*, *il10*, and *litaf* were tested in intestine samples; (iii) *npy*, *drd1b*, *drd2a*, and *drd3* were tested in brain samples. Amplification products were sequenced, and homology was verified. Gene transcript expression alterations among experimental groups are reported as relative mRNA abundance (arbitrary units). The qPCR data were processed using iQ5 optical system software version 2.0 (Bio-Rad), including GeneEx Macro iQ5 Conversion and genex Macro iQ5 files. 

### 2.10. Total Lipid Analysis

Three juveniles per tank (in triplicate, 9 per experimental group) were collected at 60 dpf; then, they were manually cut, pooled together, and divided into two parts which were subdued to the lipid extraction using chloroform:methanol (2:1) as solvents following the method proposed by Folch et al. [56]. Once extracted, the amount of total lipids was gravimetrically quantified, and the results were expressed as g/100 g of whole fish. Total lipids were also extracted from the zebrafish larvae (15 larvae each group) following the same procedure described above.

### 2.11. Statistical Analyses

Data from each analysis were checked for normality (Shapiro–Wilk test) and homoscedasticity (Levene’s test). The behavioral preference data were analyzed with a one-way analysis of variance (ANOVA) on the entire data set followed by one-sample *t*-tests (against random choice: 50%) for each individual flavor. All the data of the following experiments were then analyzed through ANOVA followed by Tukey’s multiple comparison post hoc test, performed using the software package Prism 8 (GraphPad software version 8.0.2, San Diego, CA, USA). Significance was set at *p* < 0.05.

## 3. Results

### 3.1. Behavioral Preference Test

Behavioral preference test data were initially analyzed by comparing the index across all the flavor tested. This analysis revealed that the percentage of time spent by fish close to the stimulus varied significantly according to the flavor (ANOVA, F_39,724_ = 2.42, *p* ≤ 0.0001, ${ƞ}_{g}^{2}$ = 0.12). This indicated that the subjects responded differently to the different flavors administered.

To further investigate this differential response, we analyzed separately the preference for each flavor. Since in case of no preference, the index was expected to assume a value close to 50% (see Section 2), we compared the index of each flavor against this value using one-sample *t*-tests. The comparison was performed both considering the overall experiment (0–15 min) and the 0–5 min time block separately, given the evidence of behavioral changes across the testing time. Two of the flavors showed an attractive effect on zebrafish and were used as attractive flavors in the following stages of the study. Zebrafish exposed to flavor F25 spent significantly more time close to the flavor than to the control sponge, especially at the beginning of the test (index of preference for the flavor in the initial 5 min: 59.21 ± 17.85%; one-sample *t* test: t_18_ = 2.250; *p* = 0.0372). Moreover, flavor F35 determined a long-lasting attraction during the whole experimental time (60.80 ± 11.30%; t_18_ = 4.165; *p* = 0.0006). Conversely, one of the flavors determined a repulsive effect: subjects exposed to flavor F32 spent significantly less time close to the sponge with the flavor than close to the control one (42.50 ± 9.87%; t_18_ = −3.314; *p* = 0.0039). F32 was thus identified as repulsive flavor to be used in the following part of the study.

### 3.2. Biometry

As regard survival rate, no significant differences were evident among the experimental groups at both the larval (87 ± 3, 85 ± 6, 85 ± 4, 86 ± 6, 85 ± 6, and 85 ± 5% for CTRL, PG, A1^+^, A2^+^, ROT, and A^−^, respectively) and juvenile (84 ± 4, 83 ± 5, 84 ± 5, 82 ± 4, 83 ± 6, and 83 ± 5% for CTRL, PG, A1^+^, A2^+^, ROT, and A^−^, respectively) stages.

Figure 1 shows the specific growth rate of both larvae and juveniles fed on the different diets. Considering larvae (Figure 1a), no significant differences were detected among all the groups fed the flavor-added diets (11. 63 ± 0.85, 11.47 ± 1.95, 8.95 ± 1.94, and 8.26 ± 1.31% for the A1^+^, A2^+^, ROT, and A^−^ groups, respectively) and the CTRL and PG (9.15 ± 2.82 and 8.63 ± 1.90%) groups. However, larvae fed A1^+^ and A2^+^ diets were characterized by a significantly (*p* < 0.05) higher SGR value compared to those fed the A^−^ one.

As regards juveniles (Figure 1b), no significant differences were detected between the CTRL and PG groups (9.08 ± 0.16 and 8.95 ± 0.17%). Fish fed A1^+^, A2^+^, and ROT diets showed a significantly (*p* < 0.05) higher SGR% (9.46 ± 0.31, 10.45 ± 0.31, and 9.77 ± 0.30% for the A1^+^, A2^+^, and ROT groups, respectively) compared to those fed CTRL and PG diets. Finally, the A^−^ group (9.22 ± 0.38%) did not evidence significant differences compared to the CTRL group.

### 3.3. Feed Intake

Figure 2 shows the results of the feed intake experiment conducted on zebrafish juveniles fed the experimental diets. No significant differences were detected between the CTRL and PG groups. Fish fed the A1^+^ and A2^+^ diets showed a higher feed intake with respect to both the CTRL and PG groups. Finally, no significant differences were evident among the CTRL, PG, ROT, and A^−^ groups. 

### 3.4. Histology

Considering intestine, no significant alterations in the tissue’s architecture or signs of inflammation were evident in either larvae or juveniles belonging to all the experimental groups (Figure 3a,b and Figure 3d,e, respectively). In addition, as reported in Table 2, no significant differences in all the histopathological indexes analyzed were detected in either larvae or juveniles.

As regards liver, both larvae and juveniles belonging to all the experimental groups showed a modestly fat liver parenchyma, with a diffuse presence of hepatocytes with cytoplasm filled with fat (Figure 3c,f). These results were consistent with the statistical quantification of the fat fraction percentage calculated on liver sections obtained from both larvae and juveniles, which did not evidence significant differences among the experimental groups (larvae: 47.6 ± 1.6, 49.9 ± 4.3, 42.6 ± 1.0; 51.9 ± 3.5, 46.1 ± 4.1, and 42.0 ± 5.9 for the CTRL, PG, A1^+^, A2^+^, ROT, and A^−^ groups, respectively; juveniles: 52.0 ± 4.3, 54.2 ± 4.3, 54.6 ± 4.1; 53.4 ± 3.6, 54.6 ± 1.0, and 54.7 ± 2.6 for the CTRL, PG, A1^+^, A2^+^, ROT, and A^−^ groups, respectively).

### 3.5. Real-Time PCR Results

*Growth.* Considering larvae, no significant differences were observed among the experimental groups in both *igf1* and *mstnb* gene expression (Figure 4a,b). 

As regards juveniles, no significant differences for either of the genes analyzed were detected between the CTRL and PG groups. However, a significant (*p* < 0.05) *igf1* upregulation and a significant (*p* < 0.05) *mstnb* downregulation were observed in the A1^+^ and A2^+^ groups compared to the all the other experimental groups, which did not show significant differences among them for either of the genes analyzed (Figure 4c,d). 

*Appetite.* As regards larvae, no significant differences were detected among all the experimental groups for any of the genes analyzed (Figure 5a–c). 

Considering juveniles, no significant differences were evident in *ghrl* expression between the CTRL and PG groups. Fish fed A1^+^, A2^+^, and ROT diets showed a significantly (*p* < 0.05) lower *ghrl* expression compared to those fed the CTRL one. Finally, the A^−^ group did not show significant differences in terms of *ghrl* expression compared to both the CTRL and PG groups (Figure 5d).

Considering *npy* relative gene expression (Figure 5e), the PG group showed a significantly (*p* < 0.05) lower value compared to the CTRL one. The A1^+^ group was characterized by a significant (*p* < 0.05) gene expression downregulation, while the A2^+^ group showed a significant upregulation compared to the CTRL group. Finally, both the ROT and A^−^ groups were characterized by a significant (*p* < 0.05) *npy* downregulation compared to the CTRL one. Ultimately, as regards the *lepa* relative gene expression (Figure 5f), no significant differences were detected between the CTRL and PG groups. Fish fed the A1^+^, A2^+^, and ROT diets evidenced a significant (*p* < 0.05) *lepa* upregulation compared to those fed the CTRL and PG ones. The A^−^ group showed a significantly (*p* < 0.05) higher gene expression compared to the PG group, while no significant differences were detected compared to the CTRL group.

*Reward system.* As regards the larval stage, no significant differences were detected among the experimental groups for all the genes analyzed (Figure 6a–c).

As regards the juvenile stage, CTRL and PG groups did not show significant (*p* < 0.05) differences in *drd1b* gene expression (Figure 6d). The A1^+^ group was characterized by a significant (*p* < 0.05) *drd1b* downregulation, while the A2^+^ group showed a significant (*p* < 0.05) *drd1b* upregulation compared to CTRL group. In addition, the ROT and A^−^ groups did not evidence significant differences compared to the CTRL and PG ones.

Considering the *drd2a* relative gene expression (Figure 6e), the PG group showed a significant (*p* < 0.05) upregulation compared to the CTRL one. The A1^+^ group did not show significant differences compared to CTRL, while the A2^+^ and ROT groups showed a significant (*p* < 0.05) *drd2a* upregulation compared to both the CTRL and PG ones. In addition, no significant differences in terms of *drd2a* relative gene expression were observed between the CTRL and A^−^ groups.

Finally, considering *drd3* relative gene expression (Figure 6f), the PG group showed a significant (*p* < 0.05) upregulation compared to CTRL. No significant differences were observed between the A1^+^ and CTRL groups, while the A2^+^ and ROT groups showed a significant (*p* < 0.05) *drd3* upregulation compared to both the CTRL and PG ones. In addition, the A^−^ group did not show significant differences compared to CTRL.

*Immune and stress response.* No significant differences were detected in the expression of genes involved in immune (*il1b*, *il10*, and *litaf*) and stress (*nr3c1*) response in both larvae (Figure 7a–d) and juveniles (Figure 7e–h).

### 3.6. Total Lipid Content

The analyzed larvae contained between 0.94 g (in CTRL group) and 1.82 g (in ROT group) of total lipids in 100 g of larvae pool. The total lipid content found in 100 g of larvae was 1.16, 1.38, 1.51, and 1.52 g in PG, A2^+^, A^−^, and A1^+^, respectively. For juveniles, as reported in Figure 8, the PG group was characterized by significantly (*p* < 0.05) lower total lipid content compared to CTRL. All the groups fed the flavor-added diets (A1^+^, A2^+^, ROT, and A^−^) showed a significantly higher total lipid content compared to both the CTRL and PG groups.

## 4. Discussion

Recently, in modern aquafeed formulation, traditional protein sources like FM have been replaced by more sustainable ones, often affecting feed digestibility and palatability [57], resulting in aquafeed wastage, and economic and environmental issues [15,33]. Since the use of natural feed attractants can only partially mitigate this problem, it is crucial to identify and test the efficiency of a set of new feed attractants intended for fish diets [10,19].

In light of this, the present study assessed the potential role of synthetic flavors as feed attractants during the early development of zebrafish. Specifically, the present study tested two positive/attractive synthetic flavors (F25, which showed a short-lasting attractive effect, and F35, which showed a long-lasting attractive effect) and a repulsive synthetic flavor, all of them composed by the same basic solvent (Propylene Glycol—PG).

In the present study, both zebrafish larvae and juveniles fed the PG diet generally showed comparable results to those fed the CTRL diet, underlying, as already reported in the literature [58,59], the absence of effects related to the basic solvent used. In addition, as expected, all the tested synthetic flavors had no adverse effects on the overall welfare of either larvae or juveniles. In fact, no alterations in tissue architecture and histopathological indexes, or signs of inflammations were evident in liver and intestine and no significant differences in the expression of immune-related (*il1b*, *il10*, and *litaf*) and of stress-related (*nr3c1*) markers were detected at either developmental stage. This aspect is very important since when new aquafeed ingredients or additives are used, gut welfare [60] and stress response [61] should always be carefully analyzed.

Considering all the other results obtained in the present study, a different *scenario* was observed in zebrafish larvae and juveniles. As regards the larval stage, most of the analyses performed did not show significant differences among the experimental groups, suggesting that the exposure time (21 dpf) was probably too short. Additionally, it should also be pointed out that fish larval stages are characterized by very high growth rates and quick and deep morphological modifications [62,63] which could have reduced differences among groups. However, it should be noted that besides the less pronounced growth differences of the larval stage compared to the juvenile stage, larvae fed A1^+^ and A2^+^ diets showed a significantly higher SGR% compared to the repulsive A^−^ group, underlining that the behavioral test properly selected the flavors. 

As already mentioned, the juvenile stage provided more marked responses in relation to the tested flavors, giving important insights about their potential role as novel feed attractants. Specifically, their attractive role was confirmed by the feed intake experiment and supported by the molecular analysis. The highest feed intake obtained within 15 min experiment was observed in both zebrafish juveniles fed A1^+^ and A2^+^ diets, a result that might be of interest for the aquafeed production sector to avoid feed waste. 

On this regard, it has to be mentioned that the fish feeding behavior is moved by the appetite *stimulus* that results from the interplay between orexigenic and anorexigenic signals, both produced at the peripheral-gut and central-brain levels [23]. Within this cross-talk, ghrelin is mainly produced at the gut level, playing an essential role in stimulating feeding activity [64]. Regarding the present study, the significant *ghrl* downregulation observed in zebrafish juveniles fed A1^+^, A2^+^, and ROT diets suggested that they were in a more satiated state compared to those from the other groups. These data also fully confirmed the feed intake results at the molecular level [65].

Conversely, nonobvious results were obtained by analyzing the *npy* gene expression at the brain level. NPY, besides being expressed also in other tissues like spleen, kidney, and smooth muscle [66,67], interacts with a number of different appetite signals such as corticotropin-releasing factor (CRF), cortisol, CART, orexins, and galanin, suggesting that the NPY response is not exclusively dependent from the ghrelin signal [68]. In addition, NPY is involved in several other physiological functions in fish, like circadian rhythmicity, cardiovascular activity, psychomotor activity, and sexual behavior, which could have affected the present results [69,70,71].

Differently, considering the expression of the anorexigenic signal *lepa*, a significant upregulation, in accord with *ghrl* gene expression, was observed in zebrafish juveniles fed A1^+^, A2^+^, and ROT diets, further supporting their more satiated state. Leptin expression is known to be a strong indicator of fish energy reserve availability, being directly related to the amount of adipose tissue and being involved in the mobilization of the fat stored in the liver [22,23,24]. Despite the absence of significant differences in the hepatic lipid accumulation among the experimental groups analyzed, fish fed A1^+^, A2^+^, and ROT diets showed a higher lipid content with respect to the control group, as confirmed by the total lipid analysis.

Taken together, the above-described results suggest that both attractive flavors used in the present study promoted a higher feed intake by the A1^+^ and A2^+^ groups, which in turn determined a more satiated state of these fish, a higher lipid storage, and a higher growth, which was also confirmed by the molecular analysis. It is thus suggested that diet palatability was improved by adding the two positive flavors selected in this study, especially considering the fact that feed’s smell and taste are hedonic properties able to activate the pleasure sensation during feed ingestion [20]. In this regard, it is well known that attractive and highly palatable feed positively influences appetite regulation and feeding behavior in vertebrates through the activation of the brain reward system [27]. Within this system, the neurotransmitter dopamine and its related receptors are the main actors in promoting feed motivation [72]. Particularly, in the present study, the expression of three different dopamine receptors—Drd1b, Drd2a, and Drd3—were analyzed. The Drd1b is the homolog of the mammalian D1-like receptor type, while the Drd2a and Drd3 are homologous to the mammalian D2-like receptor type [73]. D1 and D2 receptors constitute two structurally and functionally different classes of dopamine receptors, both strongly and positively related to feed intake and feed motivation [74,75]. However, the administration of highly palatable feed over a long-term period of time can contribute to the development of reward system hyposensitivity through the reduction in the expression of the dopamine receptor D2 [75,76]. This hyposensitivity condition seems to induce a compulsive eating behavior, related to an incapacity to achieve the reward expected from the feed [77]. This mechanism has been demonstrated in mammals, but it has been poorly investigated in fish. 

In the present study, both the attractive diets did not negatively affect the fish dopaminergic activity, with the most interesting results about this aspect observed during the juvenile stage. Zebrafish juveniles fed the A2^+^ (including the long-lasting F35 flavor) and, in a lesser extent, ROT diets showed a higher expression of all the dopamine receptors analyzed, highlighting a stronger dopaminergic activity compared to the CTRL group and suggesting that the reward expected from feed was maintained throughout the whole experiment. Conversely, A1^+^ group juveniles (including the F25 short-lasting flavor) showed a dopaminergic activity comparable to CTRL, suggesting that this flavor had less effect than the A2^+^ one on feeding motivation.

As regards the ROT group, which was conceptualized to understand if changing positive flavor administration during the experiment was a good strategy to avoid adaptation to the same *stimulus* and enhancing fish feeding response, it should be pointed out that the results obtained in the present study indicated that this was not a valuable strategy. The administration of a long-lasting attractive flavor as F35, interspersed with a short-lasting one as F25, reduced the overall effectiveness of the feeding scheme, probably because F25 lost its effectiveness as feed attractant over the trial, as confirmed by molecular analysis of dopaminergic activity.

Finally, the repulsive flavor used in the present study led to results that were comparable to the CTRL group, suggesting a possible adaptation by the fish to the flavor or a potential interaction between the flavor and the feed odor. However, because of the preliminary results obtained through the behavioral test, this was used as negative control.

## 5. Conclusions

All the synthetic flavors tested did not affect the overall health of fish and, for the positive ones, confirmed their attractive role, as already demonstrated by the preliminary behavioral trial. The provision of diets containing attractive flavors resulted in an increased growth rate in both larvae and juveniles. Due to the longer exposure time, results obtained from the juvenile stage provided a clearer picture of the fish responses. In fact, zebrafish juveniles fed A1^+^ and A2^+^ diets (A1^+^ cheese odor made by mixing Propylene Glycol (PG) with aromatic chemicals trimethyamine, 2-acetylpyrazine, 2-acetylpyridine, and dimethyl sulfide; and A2^+^ caramel odor, made of PG mixed with the aromatic chemicals vanillin, maltol, cyclotene, acetoin, butyric acid, and capric acid with traces of both gamma-octalactone and gamma-esalactone, respectively) showed higher appetite stimulus, feed ingestion, and growth. However, the modulation of the brain reward system suggested the selection of the A2^+^ diet (and thus the F35 flavor) as the most valuable solution for maintaining its long-lasting effect over the whole experimental period. As regards the A^−^ group (coconut odor, made by mixing PG with the aromatic chemicals gamma-eptalactone, gamma-nonalactone, delta-esalactone, and vanillin with traces of both delta-octalactone and maltol), the negative effect evidenced during the behavioral test conducted on the flavor exclusively was mitigated by mixing the flavor with the diet. 

In conclusion, the present study provides important results about the possible use of attractive synthetic flavors for the aquafeed production, opening new sustainable and more economically valuable opportunities for the aquaculture sector. Further research has to be developed, testing the same flavors on adult fish with an emphasis on reproduction, since fish life stages may have behavioral differences related to their developmental stages. 

## Figures and Tables

**Figure 1 animals-13-03368-f001:**
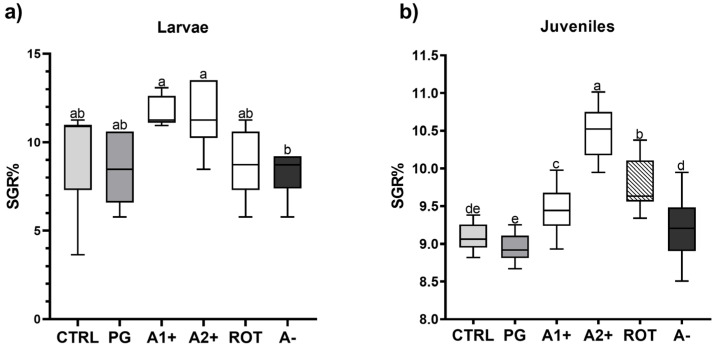
Specific growth rate (% day ^−1^) of zebrafish (**a**) larvae and (**b**) juveniles fed the experimental diets. Boxplots show minimum and maximum (whiskers), first quartile, median, and third quartile (box). ^a–e^ Different letters indicate statistically significant differences among the experimental groups (*p* < 0.05).

**Figure 2 animals-13-03368-f002:**
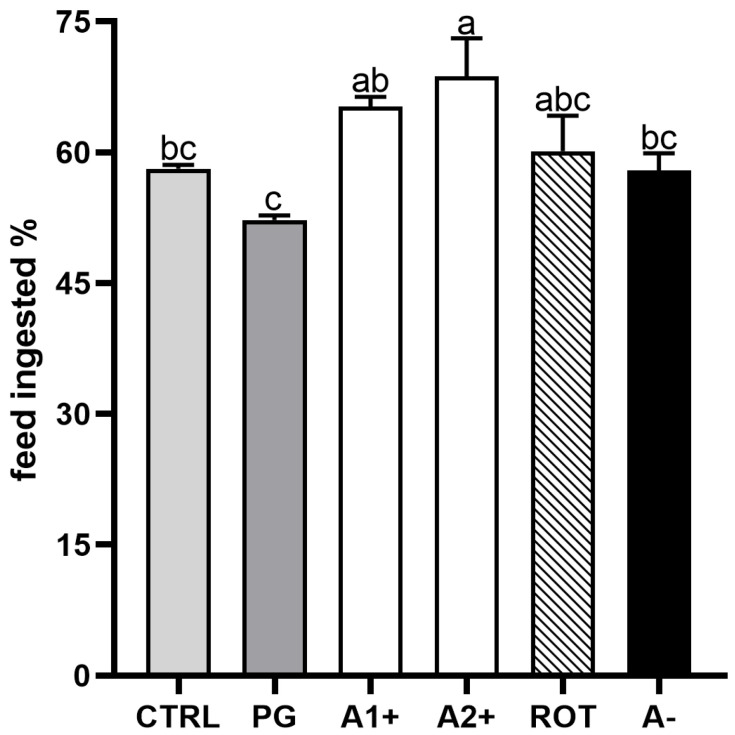
Percentage of feed ingested after 15 min administration of experimental diets in zebrafish juveniles. Results are expressed as mean + SD (*n* = 3). ^a–c^ Different letters indicate statistically significant differences among the experimental groups (*p* < 0.05).

**Figure 3 animals-13-03368-f003:**
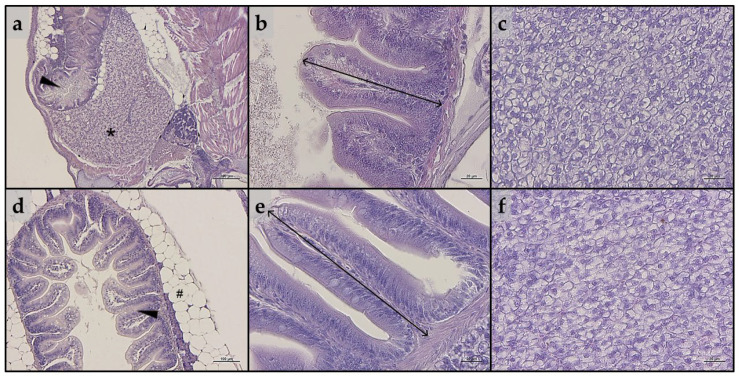
Example of histomorphology of intestine and liver parenchyma of zebrafish (**a**–**c**) larvae and (**d**–**f**) juveniles from the present study. (**a**) Representative section of liver (*) and intestine (arrowhead) of a whole-embedded zebrafish larva fed CTRL diet; (**b**) details of mucosal folds of intestine from a zebrafish larva fed PG diet (double-headed arrow indicate mucosal folds height); (**c**) liver parenchyma from a zebrafish larva fed A1^+^ diet; (**d**) section of intestine (arrowhead) from a zebrafish juvenile fed A2^+^ diet (# indicate perivisceral adipose tissue); (**e**) details of mucosal folds of intestine from a zebrafish juvenile fed ROT diet (double-headed arrow indicate mucosal folds height); (**f**) liver parenchyma from a zebrafish juvenile fed A^−^ diet. Scale bars: a = 200 μm; b,c,e,f = 20 μm; d = 100 μm.

**Figure 4 animals-13-03368-f004:**
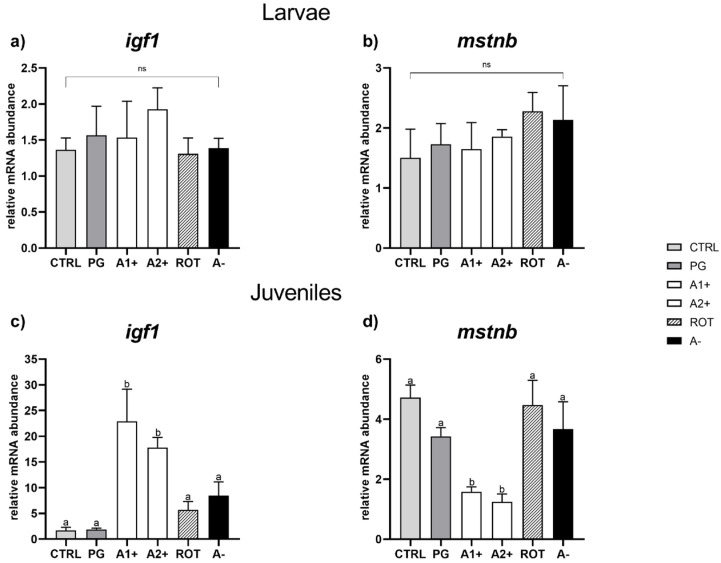
Relative mRNA abundance of genes involved in growth analyzed in whole larvae or in liver samples from juveniles. (**a**) *igf1* and (**b**) *mstnb* in larvae; (**c**) *igf1* and (**d**) *mstnb* in juveniles. Results are expressed as mean + SD (*n* = 5). ^a,b^ Different letters denote statistically significant differences among the experimental groups; ns, no significant differences.

**Figure 5 animals-13-03368-f005:**
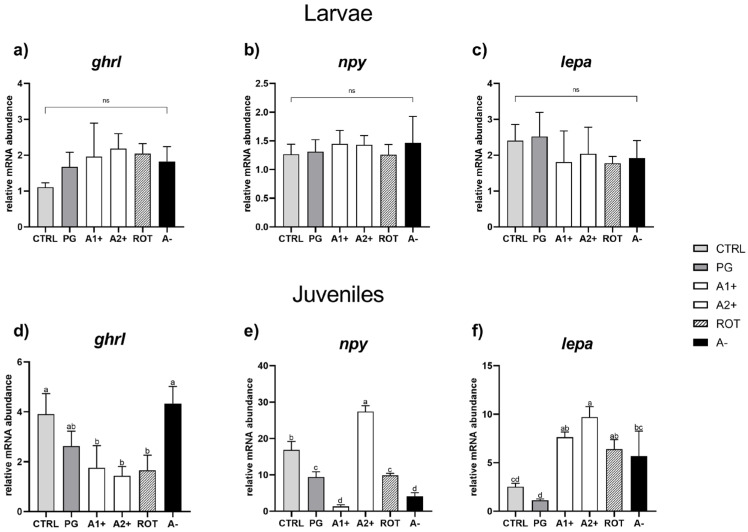
Relative mRNA abundance of genes involved in appetite regulation analyzed in whole larvae or in intestine (*ghrl*), brain (*npy*), and liver (*lepa*) samples from juveniles. (**a**) *ghrl*, (**b**) *npy*, and (**c**) *lepa* in larvae; (**d**) *ghrl*, (**e**) *npy*, and (**f**) *lepa* in juveniles. Results are expressed as mean + SD (*n* = 5). ^a–d^ Different letters denote statistically significant differences among the experimental groups; ns, no significant differences.

**Figure 6 animals-13-03368-f006:**
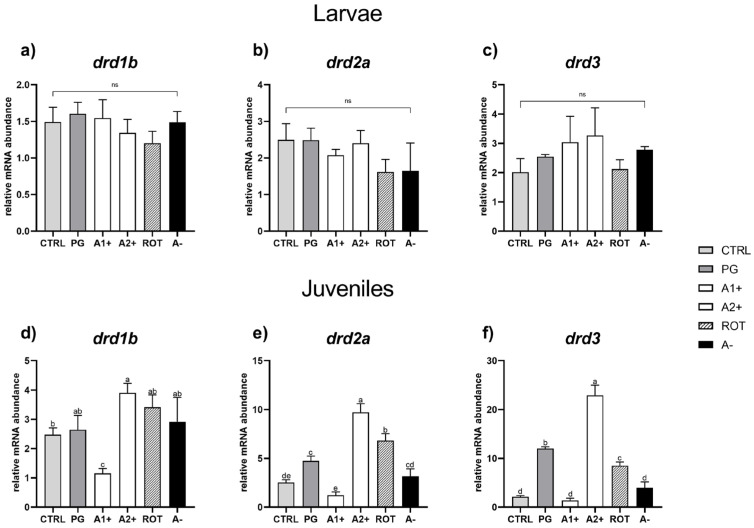
Relative mRNA abundance of genes involved in reward system analyzed in whole larvae or in brain samples from juveniles. (**a**) *drd1b*, (**b**) *drd2a*, and (**c**) *drd3* in larvae; (**d**) *drd1b*, (**e**) *drd2a*, and (**f**) *drd3* in juveniles. Results are expressed as mean + SD (*n* = 5). ^a–e^ Different letters denote statistically significant differences among the experimental groups; ns, no significant differences.

**Figure 7 animals-13-03368-f007:**
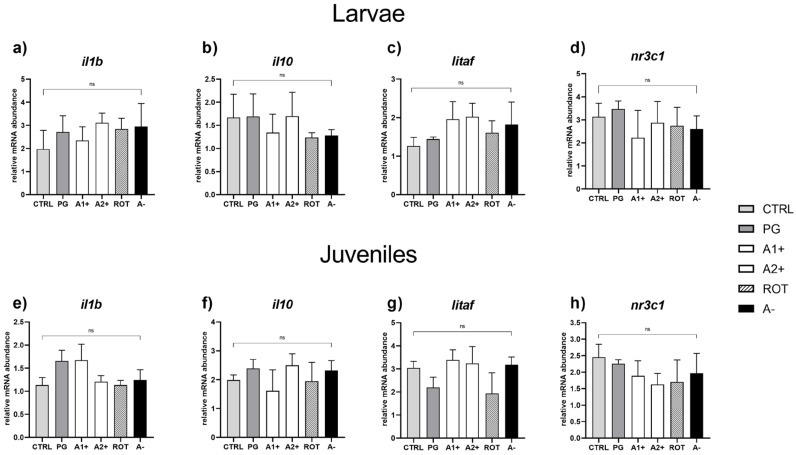
Relative mRNA abundance of genes involved in immune and stress response analyzed in whole larvae or intestine (immune response: *il1b*, *il10*, and *litaf*) and liver (stress response: *nr3c1*) samples from juveniles. (**a**) *il1b*, (**b**) *il10*, (**c**) *litaf*, and (**d**) *nr3c1* in larvae; (**e**) *il1b*, (**f**) *il10*, (**g**) *litaf*, and (**h**) *nr3c1* in juveniles. Results are expressed as mean + SD (*n* = 5). ns, no significant differences.

**Figure 8 animals-13-03368-f008:**
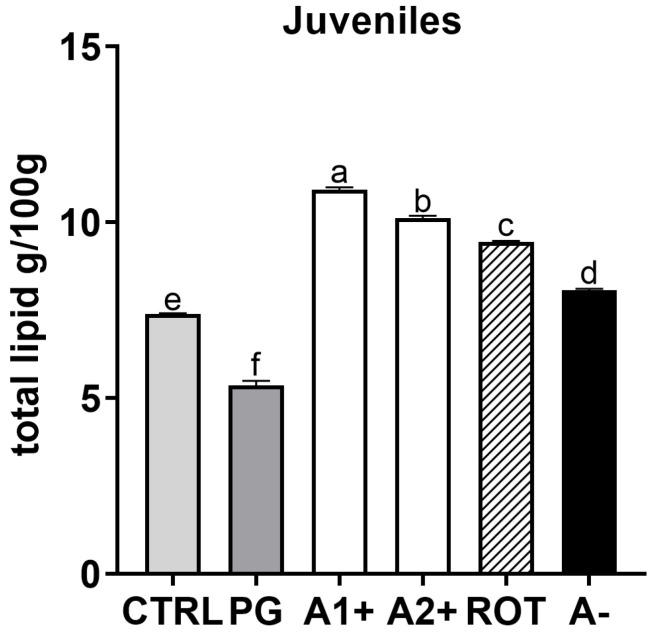
Total lipid content (g/100 g) of zebrafish juveniles. ^a–f^ Different letters show statistically significant differences among experimental groups (*p* < 0.05). Values are reported as mean + SD (*n* = 3).

**Table 1 animals-13-03368-t001:** Sequences, annealing temperature (AT), and accession number of primers used in the present study.

Gene	Forward Primer (5′-3′)	Reverse Primer (5′-3′)	AT (°C)	ID
*igf1*	GGCAAATCTCCACGATCTCTAC	CGGTTTCTCTTGTCTCTCTCAG	53	ZDB-GENE-010607-2
*mstnb*	GGACTGGACTGCGATGAG	GATGGGTGTGGGGATACTTC	58	ZDB-GENE-990415-165
*ghrl*	CAGCATGTTTCTGCTCCTGTG	TCTTCTGCCCACTCTTGGTG	58	ZDB-GENE-070622-2
*npy*	GTCTGCTTGGGGACTCTCAC	CGGGACTCTGTTTCACCAAT	60	ZDB-GENE-980526-438
*lepa*	CTCCAGTGACGAAGGCAACTT	GGGAAGGAGCCGGAAATGT	60	ZDB-GENE-081001-1
*drd1b*	CTGCGACTCCAGCCTTAATC	AGATGCGGGTGTAAGTGACC	58	ZDB-GENE-070524-2
*drd2a*	TGGTACTCCGGAAAAGACG	ATCGGGATGGGTGCATTTC	58	ZDB-GENE-021119-2
*drd3*	ATCAGTATCGACAGGTATACAGC	CCAAACAGTAGAGGGCAGG	60	ZDB-GENE-021119-1
*nr3c1*	AGACCTTGGTCCCCTTCACT	CGCCTTTAATCATGGGAGAA	58	ZDB-GENE-050522-503
*il1b*	GCTGGGGATGTGGACTTC	GTGGATTGGGGTTTGATGTG	54	ZDB-GENE-040702-2
*il10*	ATTTGTGGAGGGCTTTCCTT	AGAGCTGTTGGCAGAATGGT	56	ZDB-GENE-051111-1
*litaf*	TTGTGGTGGGGTTTGATG	TTGGGGCATTTTATTTTGTAAG	53	ZDB-GENE-040704-23
*rpl13*	TCTGGAGGACTGTAAGAGGTATGC	AGACGCACAATCTTGAGAGCAG	59	ZDB-GENE-031007-1
*arpc1a*	CTGAACATCTCGCCCTTCTC	TAGCCGATCTGCAGACACAC	60	ZDB-GENE-040116-1

**Table 2 animals-13-03368-t002:** Histological indexes measured in the intestine of larvae and juveniles fed the experimental diets.

	CTRL	PG	A1^+^	A2^+^	ROT	A^−^
**Larvae**						
Mucosal fold height	80.7 ± 0.6	93.0 ± 7.2	84.5 ± 5.5	95.4 ± 7.5	92.5 ± 7.2	75.0 ± 4.5
Inflammatory influx	+	+	+	+	+	+
Mucosal fold fusion	+	+	+	+	+	+
**Juveniles**						
Mucosal fold height	166.2 ± 23.1	168.0 ± 23.0	160.0 ± 12.0	175.3 ± 26.2	162.4 ± 12.7	164.7 ± 25.5
Inflammatory influx	+	+	+	+	+	+
Mucosal fold fusion	+	+	+	+	+	+

Data of mucosal fold height are reported as mean ± SD (*n* = 15). No significant differences were detected among experimental groups (*p* > 0.05).

## Data Availability

The data presented in this study are available on request from the corresponding author.

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
