# Peer review of "The Application of Synthetic Flavors in Zebrafish (Danio rerio) Rearing with Emphasis on Attractive Ones: Effects on Fish Development, Welfare, and Appetite"

_animals, 2023, doi:10.3390/ani13213368_

Round 1
Reviewer 1 Report
Comments and Suggestions for Authors
The main problem I have with this study is that the substances tested are not disclosed. There is no way from the information provided to know what actually was tested, so it becomes a commercial test and not a scientific one. I understand that the products are industrial secrets (intellectual property), but this doesn't help the reader to be informed as to what the scientific findings of the study are. Attractants are well known to improve feed intake and are commonly used to overcome the effects of use of unpalatable but nutritious ingredients.
Other comments:
- For the initial screening test, was each test conducted using a fresh tank? If not, how were the tanks cleaned between uses to ensure that there was no lingering of odors between substances?
- What was the experimental design of the growth trials; completely randomized, randomized complete block, etc.?
- The authors indicate that they measured survival of the fish, but I couldn't see any information about the results. The authors should provide this.
Author Response
REVIEWER 1
The main problem I have with this study is that the substances tested are not disclosed. There is no way from the information provided to know what actually was tested, so it becomes a commercial test and not a scientific one. I understand that the products are industrial secrets (intellectual property), but this doesn't help the reader to be informed as to what the scientific findings of the study are. Attractants are well known to improve feed intake and are commonly used to overcome the effects of use of unpalatable but nutritious ingredients.
Mant thanks for this suggestion. The reviewer is right and now the chemical composition of the flavors and the final odor have been included into the text.
However, to protect the intellectual property of the company the specific chemicals listed have been reported from the most concentrated to the less concentrated one, but without reporting the exact amount. We hope that this will be sufficient for the reviewer.
Other comments:
- For the initial screening test, was each test conducted using a fresh tank? If not, how were the tanks cleaned between uses to ensure that there was no lingering of odors between substances?
40 different tanks were used according to the 40 synthetic flavors tested. In this way we avoided possible contamination of water among the flavors tested. In addition, to avoid lingering of the same flavor, after each test tanks were completely washed with distilled water and refilled, for each specimen tested with the same odor. A sentence has been added in the 2.3 section.
- What was the experimental design of the growth trials; completely randomized, randomized complete block, etc.?
For fish growth performances, a completely randomized experimental design was applied. In particular, as reported in the 2.6 section, 20 randomly selected zebrafish larvae, and 20 randomly selected zebrafish juveniles were measured at each sampling point.
- The authors indicate that they measured survival of the fish, but I couldn't see any information about the results. The authors should provide this.
The reviewer is right, we didn’t report the survival rate data for our mistake. The results for both larvae and juveniles have been added at the beginning of the 3.2 section.
Reviewer 2 Report
Comments and Suggestions for Authors
1. In the Author Contribution section (LL.603-607), there appears to be an inconsistency. The author A.C.L.M. is listed in the author list, but its specific contribution is not mentioned in this section. Please clarify A.C.L.M.'s contribution in this part of the manuscript.
2. M/M & Results: The description and analysis of the behavioral preference test appear to be overly simplistic and vague. To support the claimed preference effects (either attractive or repulsive) of flavors F25, F32, and F35, it's essential to provide a more detailed and comprehensive analysis. It's unclear where the 50% threshold used for comparison originates and whether it adequately represents a 'neutral' preference effect by the 37 other flavors. To strengthen the study's validity, consider comparing the preference scores for these flavors against scores calculated from all 37 other flavors collectively or individually, not just the arbitrary value of 50%. This would provide a more robust basis for evaluating the suitability of these flavors for the study.
3. M/M: The manuscript should provide a clearer explanation of the detection threshold of additive flavors by zebrafish larvae and juveniles. This information is crucial for understanding the sensitivity of the test subjects.
4. M/M: I recommend including some other flavors with a 'neutral' preference effect as controls in the study. If it’s not feasible, it would be helpful to provide a more detailed rationale for the exclusion of these controls to enhance the clarity of the experimental design.
5. Results: Incorporating data on the survival rate of the test subjects in each treatment would contribute to a more comprehensive assessment of their well-being. Please consider including this information to provide a more holistic view of the experimental outcomes."
6. Discussion: As the authors have already pointed out, the exclusive focus on larvae and juveniles may not align with the main objective of the entire study. To ensure a comprehensive investigation, consider discussing the potential implications and limitations of this focus, as well as whether it might be beneficial to include other life stages or provide a rationale for this particular focus.
7. Tables and Figures
a. The manuscript lacks Figures 1 and 2. Please include these figures or renumber the figures that are currently included in the manuscript to enhance the clarity of the presentation
b. In Figures 6 through 9, consider enlarging the font size for improved readability. This adjustment will help readers more easily interpret the data presented in these figures.
c. The manuscript should provide a clear legend or explanation regarding the meaning of different colors or patterns used in the bars of the graphs (Figures 6 through 9). Currently, these elements are somewhat confusing nor helpful and require clarification for the reader's benefit.
8. Minor editorial issue
a. Ensure consistency in denoting standard deviation throughout the manuscript, either as 'SD' (as in the title of Figure 4) or 'standard deviation' (as in Table 2).
b. The title of Table 2 appears to be separated. Please format it correctly
c. In LL.336, 341, 343, and 345, please ensure that the d.f. values are properly formatted as subscripts.
Author Response
REVIEWER 2
In the Author Contribution section (LL.603-607), there appears to be an inconsistency. The author A.C.L.M. is listed in the author list, but its specific contribution is not mentioned in this section. Please clarify A.C.L.M.'s contribution in this part of the manuscript.
The reviewer is right. We specified the contribution of the highlighted author in the dedicated section.
- M/M & Results: The description and analysis of the behavioral preference test appear to be overly simplistic and vague. To support the claimed preference effects (either attractive or repulsive) of flavors F25, F32, and F35, it's essential to provide a more detailed and comprehensive analysis. It's unclear where the 50% threshold used for comparison originates and whether it adequately represents a 'neutral' preference effect by the 37 other flavors. To strengthen the study's validity, consider comparing the preference scores for these flavors against scores calculated from all 37 other flavors collectively or individually, not just the arbitrary value of 50%. This would provide a more robust basis for evaluating the suitability of these flavors for the study.
The referee is right, our description of the methods was too short because the test was already described in a previously published study and it was intended to be preliminary to the main study. We now amended by adding all available details in the method section.
Now the experiment is described in the revised manuscript and we think that all necessary information is reported.
- M/M: The manuscript should provide a clearer explanation of the detection threshold of additive flavors by zebrafish larvae and juveniles. This information is crucial for understanding the sensitivity of the test subjects.
We added all the details in the method section, including the detection threshold.
- M/M: I recommend including some other flavors with a 'neutral' preference effect as controls in the study. If it’s not feasible, it would be helpful to provide a more detailed rationale for the exclusion of these controls to enhance the clarity of the experimental design.
At this stage, adding new test substances is not possible according to canonical experimental practices of randomization (Altman, 1991). The experiment was run by testing simultaneously subjects randomly selected from a large pool, and each subject was randomly assigned to one testing substance. Care was taken to test simultaneously a similar number of subjects for each testing substance. This design ensured the maximum control for stochastic effects that could affect the experiment, such as differences between fish or daily variation due to non-controllable atmospheric factors (e.g., pressure). We added these details in the new method section. In addition, the title was modified adding “with emphasis on attractive flavors’’.
As conclusion, we can state that if we add now additional stimuli, they cannot be statistically compared to the ones of the early studies because of absence of randomization (Altman, 1991), different origin of the subjects (different pool; e.g., Lange et al., 2016; Quadros et al., 2016), and potential changes in environmental conditions (e.g., Parker et al., 2012).
- Results: Incorporating data on the survival rate of the test subjects in each treatment would contribute to a more comprehensive assessment of their well-being. Please consider including this information to provide a more holistic view of the experimental outcomes."
The reviewer is right, we didn’t report the survival rate data for our mistake. As suggested also by the Reviewer 1, the results for both larvae and juveniles have been added at the beginning of the 3.2 section.
- Discussion: As the authors have already pointed out, the exclusive focus on larvae and juveniles may not align with the main objective of the entire study. To ensure a comprehensive investigation, consider discussing the potential implications and limitations of this focus, as well as whether it might be beneficial to include other life stages or provide a rationale for this particular focus.
A short sentence was added in the conclusion section.
- Tables and Figures
- The manuscript lacks Figures 1 and 2. Please include these figures or renumber the figures that are currently included in the manuscript to enhance the clarity of the presentation.
The reviewer is right, we merged some figures during the manuscript editing and we did not update the number of the figures. Now figures are corrected.
- In Figures 6 through 9, consider enlarging the font size for improved readability. This adjustment will help readers more easily interpret the data presented in these figures.
The font sizes have been increased in order to improve readability.
- The manuscript should provide a clear legend or explanation regarding the meaning of different colors or patterns used in the bars of the graphs (Figures 6 through 9). Currently, these elements are somewhat confusing nor helpful and require clarification for the reader's benefit.
The colors of histograms were randomly chosen by the authors. A legend has been added in figures from 6 to 9 to better clarify the match between colors and groups.
- Minor editorial issue
- Ensure consistency in denoting standard deviation throughout the manuscript, either as 'SD' (as in the title of Figure 4) or 'standard deviation' (as in Table 2).
All 'standard deviations have been replaced by 'SD'.
- The title of Table 2 appears to be separated. Please format it correctly
The title is not separated but is reported according to MDPI guidelines. In particular, above the table is reported the description while under the table technical information about data are reported.
- In LL.336, 341, 343, and 345, please ensure that the d.f. values are properly formatted as subscripts.
Thank you for the suggestions. The only d.f. value was properly formatted in the revised manuscript. The other 'F' values (F25, F35, and F32) in the 3.1 section were referred to the flavors’ codes.
Round 2
Reviewer 2 Report
Comments and Suggestions for Authors
Thank you for addressing most of the comments and suggestions from this reviewer. My only additional comment is regarding the bar graph figures: please use "+SD" instead of "±SD" to accurately represent the error bars.
Author Response
Thank you for addressing most of the comments and suggestions from this reviewer. My only additional comment is regarding the bar graph figures: please use "+SD" instead of "±SD" to accurately represent the error bars.
Mant thanks, this has been corrected throughout the text